# An Occurrence and Exposure Assessment of Paralytic Shellfish Toxins from Shellfish in Zhejiang Province, China

**DOI:** 10.3390/toxins15110624

**Published:** 2023-10-24

**Authors:** Qin Weng, Ronghua Zhang, Pinggu Wu, Jiang Chen, Xiaodong Pan, Dong Zhao, Jikai Wang, Hexiang Zhang, Xiaojuan Qi, Xiaoli Wu, Junde Han, Biao Zhou

**Affiliations:** 1School of Public Health, Hangzhou Medical College, Hangzhou 310013, China; 881012022119@hmc.edu.cn; 2Zhejiang Provincial Center for Disease Control and Prevention, Hangzhou 310051, China; rhzhang@cdc.zj.cn (R.Z.); pgwu@cdc.zj.cn (P.W.); jchen@cdc.zj.cn (J.C.); xdpan@cdc.zj.cn (X.P.); dzhao@cdc.zj.cn (D.Z.); jkwang@cdc.zj.cn (J.W.); hxzhang@cdc.zj.cn (H.Z.); xjqi@cdc.zj.cn (X.Q.); xlwu@cdc.zj.cn (X.W.); 3Department of Epidemiology and Health Statistics, School of Public Health, Faculty of Medicine, Hangzhou Normal University, Hangzhou 311121, China; 2021112012151@stu.hznu.edu.cn

**Keywords:** paralytic shellfish toxins, saxitoxin, dietary exposure, risk assessment

## Abstract

The intake of paralytic shellfish toxins (PSTs) may adversely affect human health. Therefore, this study aimed to show the prevalence of PSTs from commercially available shellfish in Zhejiang Province, China, during the period of frequent red tides, investigate the factors affecting the distribution of PSTs, and assess the risk of PST intake following the consumption of bivalve shellfish among the Zhejiang population. A total of 546 shellfish samples were collected, 7.0% of which had detectable PSTs at concentrations below the regulatory limit. Temporal, spatial, and interspecific variations in the occurrence of PSTs were observed in some cases. The dietary exposure to PSTs among the general population of consumers only was low. However, young children in the extreme scenario (the 95th percentile of daily shellfish consumption combined with the maximum PST concentration), defined as 89–194% of the recommended acute reference doses, were possibly at risk of exposure. Notably, *Arcidae* and mussels were the major sources of exposure to toxins. From the public health perspective, PSTs from commercially available shellfish do not pose a serious health risk; however, more attention should be paid to acute health risks, especially for young children, during periods of frequent red tides.

## 1. Introduction

In recent decades, harmful algal blooms (HABs) have occurred widely and frequently in coastal areas worldwide, adversely impacting marine ecosystems. Paralytic shellfish toxins (PSTs) are the most common toxins associated with seafood toxin syndromes [1]. PSTs can be produced by the accumulation of certain marine dinoflagellates in filter-feeding bivalve mollusks, such as scallops, mussels, oysters, and clams [2]. They are a large family of neurotoxins, including saxitoxin (STX) and over 57 currently known STX analogs [3]. For each PST, its toxicity is usually expressed as STX equivalents (STX eq.) [4,5]. Common PSTs are generally divided into three groups based on R4 side-chain: carbamate toxins (STX, neosaxitoxin [neoSTX], and gonyautoxins [GTX]1–4), decarbamoyl (dc) toxins (dcSTX, dcneoSTX, and dcGTX1-4), and N-sulfocarbamoyl toxins (GTX5-6 and C1-4), with toxicity decreasing sequentially [6]. The ingestion of large amounts of PSTs may interfere with the function of the voltage-gated sodium channel, resulting in a severe and fatal illness known as paralytic shellfish poisoning (PSP) [7,8]. It poses a threat to public health and results in economic losses [3,4]. In fatal cases, victims suffer respiratory arrest 2–12 h after consuming shellfish contaminated with PSTs [4]. Currently, there is no specific treatment for PSTs, and supportive care has been the primary treatment [9,10].

Owing to the impact of PSTs on seafood safety and human health, shellfish contamination with PSTs has become a global public health concern. To ensure that shellfish are safe for consumption, many organizations in various countries, including the U.S. Food and Drug Administration and Health Canada, have adopted a regulatory limit of 800 µg of STX equivalents/kg shellfish meat [4,11,12,13,14]. The European Union updated STX-diHCL equivalents as the unit of measure for PSP in 2021 [15]; however, the previous unit remains more widely used in recent studies on PST levels and risk assessments worldwide [5,14,16,17,18,19,20]. Considering the potential health risks of PSTs, the European Food Safety Authority (EFSA) recommended an acute reference dose (ARfD) of 0.5 µg STX eq./kg b.w. for PSTs [4], whereas the Joint Food and Agriculture Organization of the United Nations/World Health Organization/Intergovernmental Oceanographic Commission of United Nations Educational, Scientific and Cultural Organization (FAO/WHO/IOC) ad hoc Expert Consultation on Biotoxins in Bivalve Molluscs recommended an ARfD of 0.7 µg STX eq./kg b.w. [21].

PSTs can be transported and bioaccumulated in shellfish through the food chain before being consumed by humans [2]. Previous studies have shown that shellfish contamination with PSTs is a major health challenge in many coastal areas, such as Malaysia [22], Tasmania [23], India [24], and Venezuela [25]. Recently, high PST levels exceeding regulatory limits have been detected in shellfish from southeastern China, affecting seafood safety for consumers [19]. Numerous studies have revealed that PST levels and detection rates highly depend on the sampling times, sites [19,26], and shellfish species [18,27,28]. Therefore, it is challenging to accurately predict PST contamination in seafood, which further increases the risk to human health.

To protect human health, various technologies have been applied to reduce PST contamination in samples, but no method has been shown to effectively eliminate PSTs [29], which results in either the incomplete elimination of PSTs or the generation of toxic by-products. For example, PSTs are heat-stable; therefore, home processing (cooking and steaming) usually fails to destroy PSTs completely when eating chowder [4], and cooking cannot change the high bioaccessibility of PSTs in shellfish [6]. The removal of PSTs using chlorination is highly pH-dependent [30] and increases the risk of generating disinfection by-products [31]. Probiotic Lactic Acid Bacteria can eliminate some, but not all, PSTs from samples [32]. Considering that none of these approaches can eliminate PSTs from samples completely and safely, further studies focusing on monitoring the PSTs in food and assessing the risk of PST exposure to consumers are urgently needed.

Therefore, it is important to study the human dietary exposure to PSTs. However, only a few studies have evaluated the risk of exposure to PSTs among populations [20,33,34,35,36,37]. In addition, some dietary studies might not have fully represented local trends, because non-local consumption estimates were used [33,36] and a small number of toxin types in a small sample size were included [34], which may have led to some uncertainty. Notably, some results have shown that people were exposed to acceptable PST levels [34], meanwhile, people in some areas might have been exposed to high PST levels [20,35], especially during the HAB period [35]. Zhejiang Province, located in the eastern coastal area of China, is a major marine fishery province and an area with frequent red tides, where local shellfish may be at risk of PST contamination. However, to the best of our knowledge, only one relevant study has been conducted in Ningbo, Zhejiang Province, showing that the population was at risk of PSP during the HAB period [35]. Therefore, this study aimed to monitor the PST levels in commercially available shellfish in Zhejiang Province, explore the factors affecting the distribution of PSTs, and assess the risk of exposure to PSTs among Zhejiang consumers during periods of frequent red tides using shellfish consumption data from local residents in Zhejiang Province.

## 2. Results

### 2.1. PST Components in Zhejiang Province

The detection rate of PSTs in the samples was 7.0% (38/546 samples). Of these positive samples, 47.4% (18/38) were contaminated with two or more toxins, with a maximum of five types of PSTs contaminating *Scapharca subcrenata* at the same time. Eight types of PSTs were detected in the shellfish samples: STX, neoSTX, dcGTX3, dcGTX2, GTX3, dcSTX, GTX2, and GTX5. However, the composition of PSTs in the samples differed according to the time of the year. All eight types of PSTs were detected in 2019, whereas STX, GTX2, GTX3, and GTX5 were detected in 2018. The dominant toxin was GTX5 (detection rate: 4.2%), followed by STX (2.4%) and dcGTX2 (2.4%). The remaining analogs were present at relatively low levels, with a detection rate of approximately 0.2–1.5% (Table 1). Among the different individual PSTs detected, GTX5 had the highest raw concentration, followed by STX, GTX2, neoSTX, GTX3, dcGTX2, dcSTX, and dcGTX3.

### 2.2. Factors Affecting the Distribution of PSTs

The results of the sample analyses, including the sample sizes, detection rates, and PST concentrations in shellfish according to different subgroups, are presented in Table 2. Only two *Atrina pectinata* specimens were collected because of the low culture production in Zhejiang Province [38]. The non-detected (ND) results are presented as 0 and limit of detection (LOD) to obtain the lower bound (LB) and upper bound (UB) of the concentrations, respectively [39,40]. Two samples had relatively high toxin levels but different toxin profiles after the data processing, with concentrations ranging from 186.9 (LB) to 291.9 (UB) µg STX eq./kg for *Scapharca subcrenata* and from 217.4 (LB) to 278.2 (UB) µg STX eq./kg for *Arcidae* (except *Scapharca subcrenata*). In the present study, the former concentration was used as the maximum level in the ranking order of the UB.

To better understand the differences in the PST detection rates among the shellfish species, Fisher’s exact test was used to check for statistically significant differences. The positive detection rates of the PSTs in *Atrina pectinate*, *Scapharca subcrenata*, *Arcidae* (except *Scapharca subcrenata*), oysters, scallops, and mussels were 0, 41.7%, 5.7%, 0, 2.8%, and 6.8%, respectively. A significant difference in detection rates was observed among shellfish species (*p* < 0.001). Bonferroni correction showed that the detection rate of the PSTs was significantly higher in *Scapharca subcrenata* than in the other detectable groups (*p* < 0.05). In addition, *Scapharca subcrenata* had relatively higher concentrations than the other species.

These samples were also categorized according to sampling time: May–September 2018 and May–September 2019. Both the samples collected in August 2018 and August 2019 showed no detectable PSTs. The detection rates of the PSTs were significantly higher in June, July, and September 2019 than in the corresponding months in 2018 (*p* < 0.05). In the present study, September 2019 had the highest detection rate, followed by July 2019, with relatively high concentrations among the samples.

The shellfish samples were also classified according to sampling sites: Hangzhou, Ningbo, Taizhou, Wenzhou, and Zhoushan. The results of the Fisher’s exact test showed that the incidence of PSTs in the samples differed significantly among the sampling sites (*p* < 0.001). The highest incidence of PSTs was observed in the shellfish collected from Hangzhou (41.4%), followed by Taizhou (12.9%), Zhoushan (5.6%), and Ningbo (2.4%). The samples from Wenzhou contained no detectable toxins. Relatively high PST levels were detected in the shellfish from Zhoushan and Taizhou.

### 2.3. Dietary Exposure Assessment

The dietary exposure values and %ARfD of the PSTs in the general population of consumers only and in different age groups are presented in Table 3. In scenarios A, B, and C, the exposure to PSTs in the general population ranged from 0 to 0.20 µg STX eq./kg b.w. The acute exposure to PSTs based on a large portion size of 85.5 g/day (the 95th percentile of daily consumption for the general population of consumers only) with a maximum toxin level of 186.9 (LB)–291.9 (UB) µg STX eq./kg from *Scapharca subcrenata* was 0.28–0.43 µg STX eq./kg b.w. in Scenario D, accounting for 56%–86% of the ARfD recommended by EFSA (0.5 µg STX eq./kg b.w. [4]) and 40%–61% of that recommended by the FAO/WHO/IOC (0.7 µg STX eq./kg b.w. [21]).

In scenarios A, B, and C, the exposure to PSTs in different age groups was 0–0.44 µg STX eq./kg b.w.. However, in scenario D, the acute exposure to PSTs in different age groups was 0.21–0.97 µg STX eq./kg b.w.. Among the different age scenario groups, young children aged <6 years had the highest exposure in each scenario and were at risk in scenario D, with exposure values ranging from 0.62 to 0.97 µg STX eq./kg b.w., accounting for approximately 124–194% of the ARfD recommended by EFSA and 89–139% of that recommended by the FAO/WHO/IOC.

Regarding the contributions of different species to the acute exposure to PSTs in the general population of consumers only, *Arcidae* (except *Scapharca subcrenata*) (23.1%–40.8%), *Scapharca subcrenata* (24.2%–35.1%), and mussels (19.8%–23.7%) were the top three contributors of PSTs when using the upper or lower bounds of the concentrations (Figure 1).

### 2.4. One-Time Safe Consumption of Shellfish for Young Children

Since young children in Zhejiang are at risk of PSP, the amounts of shellfish species that could be consumed within 24 h without posing a health risk were calculated (Table 4). The calculated safe consumption levels of the edible portions of both *Arcidae* and mussels for young children were lower than the 95th percentile of daily consumption level, consistent with the results derived from scenario D. For practical purposes, the fresh weights (including shells) of the shellfish were calculated; the fresh weights of *Arcidae*, mussels, and scallops were approximately 115–160 g/d, 82–115 g/d, and 206–289 g/d, respectively.

## 3. Discussion

The prevalence of PSTs is an important public health concern. PSTs in commercially available shellfish from Zhejiang Province, China, were analyzed to ensure food safety and promote public health. A general comparison with other regions was made due to specific monitoring procedures varying across regions and the occurrence of PSTs being influenced by different factors, including marine dinoflagellates, red tides, climate, and sea surface temperature. A previous study conducted in Fujian found that the dominant toxin was GTX5 (detection rate: 4.46%), which is consistent with the results in the present study [19]. However, some other studies have reported contradictory results. A study conducted in Jiangsu showed that STX was the most frequently detected PST [41]. Furthermore, dcSTX, GTX5, GTX3, and GTX2 were the main types of PSTs in shellfish in Hainan [42]. Compared to Fujian [19] (detection rate: 10.91%, maximum concentration: 2137.10 µg STX eq./kg), Hainan [42] (15.2%, 524.5 µg STX eq./kg), and Shenzhen [20] (13.8%, 715.60 µg STX eq./kg), which are also in the southeastern coastal area of China, Zhejiang Province had a much lower detection rate (7.0%) and maximum concentration of PSTs (186.9 (LB)~291.9 (UB) µg STX eq./kg) in shellfish samples. Except for the bivalve shellfish in the southeastern coastal area of China, all bivalve shellfish were found to contain PSTs in the coastal waters of Qinhuangdao, China, with a high PST concentration reaching 607 µg STX eq./kg, which was closely associated with the sampling period at the end of the red tide due to large amounts of toxins potentially being dissolved in the seawater [14]. The PST concentrations in the tissues of butter clams collected in three communities in the Kodiak Islands, Alaska (the City of Kodiak, Ouzinkie, and Old Harbor), were 160–3850 µg STX eq./kg, 460–5780 µg STX eq./kg, and 410–6720 µg STX eq./kg, respectively [17], which were associated with the rapid accumulation of PSTs in shellfish and regional warming events in Alaska. These studies showed the wide range of the distribution and fluctuation of PSTs, which are associated with various factors.

In Zhejiang Province, the detection rate of PSTs was higher for *Scapharca subcrenata* than for other species at 41.7%, and this rate was slightly higher than that reported in Hainan Province [42] (33.3%) and lower than that reported in Shenzhen [20] (50%). Numerous studies have revealed that these bivalve species accumulate and eliminate PSTs differently [18,28]. Generally, oysters accumulate PSTs at a much slower rate and contain relatively lower PST levels in their bodies than mussels and scallops [18]. *Scapharca subcrenata* can accumulate PSTs relatively rapidly; however, the biotransformation process is complex and further studies on the accumulation, metabolism, and transformation of PSTs in their bodies are needed [43]. Furthermore, the viscera of shellfish usually contain high concentrations of toxins that could be poisonous when consumed [44]. Since the meat of these mollusks is usually used in food preparation, it is important to comprehensively monitor the PST levels in *Scapharca subcrenata* to minimize the risk of PSP.

PSTs were detected during May–September 2018 and May–September 2019, except in August, with higher detection rates and levels of PST contamination being observed in July and September 2019, which may have been influenced by the frequency of toxic red tides [45,46] and variations in sea surface temperature (SST) in the East China Sea. According to a document released by the Zhejiang Provincial Department of Natural Resources, *Gymnodinium catenatum* was one of the major algae that caused toxic red tides in 2018 [46]. Previous studies have found that the suitable growth temperature for *G. catenatum* isolated from the East China Sea was 20–26 °C, whereas they showed almost no growth at temperatures of <18 °C or >28 °C [47]. The mean SST in the East China Sea increased in May, peaked in August (>28.5 °C), and then declined [48], which may explain the phenomena.

Regarding sampling sites, the detection rate of PSTs in Hangzhou was relatively high (41.4%). However, to the best of our knowledge, most shellfish collected in Hangzhou originate from the Bohai Sea, including the Hebei and Shandong Provinces. PSTs were reported in Hebei Province [49] (detection rate: 4.7%, exceeding standard rate: 3.0%), five regions around the Bohai Sea [26] (98%, 12%), and Qinhuangdao [14] (100%, 0%), suggesting that PST contamination in the Bohai Sea was concerning and that the prevalence of PSTs in shellfish from the Bohai Sea fluctuated widely. Furthermore, shellfish poisoning after consumption occurred in Qinhuangdao and Tangshan in the Hebei Province between late April and early May 2019 [49], which may have contributed to this result.

To assess the health risks of PST in shellfish, a cautious approach and an exaggerated risk consistent with conservative guidelines were used in the present study. To obtain a large portion of consumption, we included valid consumer days for consumers only on bivalves when calculating the percentile of consumption, with consumer days indicated as independent observations in the database without averaging [39]. Finally, the consumption rates of the bivalves used in the present study were similar to those of some high shellfish consumers in Korea in Asia [34] and higher than those of Nha Trang City, southern coastal Vietnam [50], and Western Brittany, France [51]. This difference in consumption may be related to several factors, including the research methodology, dietary habits of the population, and geographic location.

The exposure assessment results showed that the dietary exposure to PSTs in the general population of consumers only was below the ARfDs recommended by EFSA and the FAO/WHO/IOC. However, young children are vulnerable because of their lower body weight and relatively high levels of exposure to PSTs. They were at a risk of exposure to PSTs above the recommended ARfDs in the extreme scenario (scenario D), in which the 95th percentile of daily shellfish consumption and maximum PST concentrations were used. We also analyzed the shellfish consumption data obtained using the food frequency method in another simultaneous survey, with acute dietary exposure for young children being reported as 0.10–0.15 µg STX eq./kg b.w. in scenario D, approximately 15.4% of previous results, which may be related to the tendency of the method to produce long-term consumption data rather than short-term data. In previous studies in China, children (2–7 years old) in Tangshan also had the highest dietary exposure to PSTs, at 0.7167 µg STX eq./kg b.w., in the spring of 2020 [33]. In addition, acute dietary exposure to PSTs among Shenzhen residents was 2.4–3.7 times higher than the recommended ARfDs, which was based on the 99th percentile of daily shellfish consumption and the maximum PST concentration in *Chlamys nobilis* [20]. Results from Korea showed that acute dietary exposure to PSTs was 0.3 µg STX eq./kg b.w. in the general population of consumers only [34], which was within the range of the results in the present study (scenario D: 0.28(LB)–0.43(UB) µg STX eq./kg b.w.). They also found that older individuals had the highest risk of exposure (0.32 µg STX eq./kg b.w.), followed by those in the 20–64 age group, whereas children had a relatively low risk of exposure, ranging from 0.18 to 0.30 µg STX eq./kg b.w., which was inconsistent with the results of the present study. Compared to results from the other areas, the risks of exposure to PSTs in the general population of Zhejiang residents were acceptable. However, more attention should be paid to acute health risks, especially in children.

This study had some limitations. First, using consumers only data may have overestimated the results. Second, the shellfish contamination levels were obtained from a survey in 2018–2019, whereas the consumption data were obtained from a survey in 2015–2016, which were mismatched. Third, the shellfish in this study were sampled during the high red tide season, and the concentration data may have been higher than usual. Fourth, although the main commercially available shellfish consumed in Zhejiang were surveyed, toxins were also found in marine invertebrates other than shellfish [52], which could also pose a risk to human consumption. Finally, we lacked continuous and systematic monitoring of PSTs throughout the year, as we only monitored the contamination and exposure to PSTs for a few months in 2018 and 2019. In future studies, we will strengthen the monitoring of toxins for more reliable findings. These limitations might influence the exposure of people to PSTs. However, to a certain extent, this study provides useful data on the risk profile of the dietary intake of PSTs among consumers only in Zhejiang Province, China, until more exposure assessment data are available.

## 4. Conclusions

The PST levels in each shellfish sample were relatively low, and none exceeded the regulatory limits. We found that the prevalence of PSTs was influenced by various factors, such as sampling species, sampling time, sampling site, red tide, and SST. The dietary exposure to PSTs among the general population of consumers only was within acceptable levels, whereas young children aged <6 years were possibly at a high risk of exposure to PSTs under an extreme scenario. Therefore, we calculated the quantities of different shellfish species that are safe for young children to consume at one time and suggested consuming more of other shellfish that contain lower levels of toxins instead of *Arcidae* and mussels. Local authorities should improve the screening of shellfish products before they reach the market, especially species such as *Arcidae* and mussels, during periods of frequent red tides to ensure product quality and reduce the risk of PST consumption. In addition, since shellfish is not the only dietary source of potential PST intake, a more comprehensive dietary risk assessment based on the contributions of different diets should be conducted in the future.

## 5. Materials and Methods

### 5.1. Sample and Preparation

Shellfish samples were collected from five coastal cities in Zhejiang Province, including Hangzhou, Ningbo, Taizhou, Wenzhou, and Zhoushan. The samples were collected using a random sampling method from May 2018 to September 2018 and from May 2019 to September 2019 by selecting farmers’ markets, stores, supermarkets, and other locations where residents frequently purchase shellfish. The sample size was determined using the following formula [53]:(1)$$N=\frac{Z^{2}\times \left[P\times \left(1-P\right)\right]}{d^{2}}$$
where N = sample size; Z = 1.96 for 95% confidence level; P = 0.5 for the expected percentage of samples containing toxins; and d = 10%, indicating precision. According to this formula, a minimum of 96.04 samples had to be collected. In the present study, 546 samples were collected, including *Atrina pectinata* (n = 2), *Scapharca subcrenata* (n = 24), *Arcidae* (except *Scapharca subcrenata*) (n = 53), oysters (n = 77), scallops (n = 36), and mussels (n = 354). The shellfish samples were collected in duplicate by trained investigators. The samples were transported to the laboratory in a cooler containing dry ice. The collected samples were then cleaned with water to remove contaminants. The shellfish meat was removed, homogenized, and stored at ≤−18 °C until the chemical analysis. The edible portions of the scallop samples were tested.

### 5.2. Chemical Reagents

Acetonitrile and formic acid were purchased from Merck (Darmstadt, Germany). Ammonium formate and ammonia (25–28%) were purchased from CNW Technologies GmbH (Düsseldorf, Germany). The water was distilled and purified using a Millipore water purification system (Millipore Ltd., Bedford, MA, USA). Standard PST reagents (STX, dcSTX, neoSTX, GTX1, GTX2, GTX3, GTX4, dcGTX2, dcGTX3, and GTX5) were purchased from the National Research Council of Canada (Halifax, NS, Canada).

### 5.3. Sample Analysis

Local laboratories in Zhejiang Province analyzed the PST concentrations in food products. The ten PSTs were determined in the samples, as previously described [19,49], but with some modifications. Briefly, 2 g of the homogenized specimen was extracted through vortexing with 8 mL of a 0.5% acetic acid aqueous solution and was subsequently heated in a boiling water bath. After cooling, 1 mL of the previous extract was vortexed with 5 µL of ammonia. Then, 0.25 mL of the extract was added to an activated Supelco ENVI-Carb solid-phase extraction column. After draining the fluids, it was washed and eluted using 700 µL of water and 2 mL of a 20% acetonitrile aqueous solution (containing 0.8% acetic acid), respectively. The eluent was subsequently filtered through a 0.22 µm syringe membrane and stored at 4 °C for measurement.

A Waters Xevo TQ-XS triple quadrupole mass spectrometer (Waters Corporation, Milford, MA, USA) with an electrospray ionization source was used for mass spectrometry detection. The chromatographic separation of the PSTs was performed on a TSK-Gel Amide-80 (2.0 mm × 150 mm; 5 µm) column maintained at 40 °C with a flow rate of 0.4 mL/min. Mobile phase A was an aqueous solution containing 2 mmol/L of ammonium formate and 50 mmol/L of formic acid. Mobile phase B was an acetonitrile solution containing 2 mmol/L of ammonium formate and 50 mmol/L of formic acid. The gradient elution program was initiated with 20% A. The sample injection volume was 5 µL.

### 5.4. Method Validation and Quality Control

The staff of all the participating laboratories were trained uniformly in the experimental methods. The method validation parameters of accuracy, precision, linearity, and limit of detection (LOD), etc., adopted at the local laboratories were verified before including their data in the database of contaminants. The LOD was calculated as 3× signal-to-noise ratios. The LOD of the ten individual PSTs (STX, GTX1, neoSTX, dcGTX3, GTX4, dcGTX2, GTX3, dcSTX, GTX2, and GTX5) in the present study was 20 µg/kg.

### 5.5. Contamination Data Processing

The PST content in each sample was calculated using the following formula [54]:(2)$$STX\;eq.=\sum_{i=1}^{n}x_{i}\cdot r_{i}$$
where $x_{i}$ is the content of individual PSTs and $r_{i}$ is the toxicity equivalency factors (TEF) of the PSTs. According to GB 5009.213-2016, the TEFs for STX, neoSTX, GTX1, GTX2, GTX3, GTX4, GTX5, dcSTX, dcGTX2, and dcGTX3 were 1, 0.92, 0.99, 0.36, 0.64, 0.73, 0.06, 0.51, 0.65, and 0.75, respectively [54].

A substitution method was applied to deal with left-censored data [39,40]. In the present study, 0 and LOD were used for ND results; in the lower-bound (LB) scenario, all ND results were set to 0, whereas, in the upper-bound (UB) scenario, ND results were set to the LOD of each toxin [55].

### 5.6. Food Consumption Data

Food consumption data were obtained from the Zhejiang Food Consumption Survey conducted between 2015 and 2016 across 10 cities in Zhejiang Province using three non-consecutive, 24 h dietary recall face-to-face interviews conducted on 1 weekend day and 2 weekdays, with at least 5 days between the two adjacent surveys. Individual body weights were simultaneously measured. Approximately 19,968 residents (living for ≥6 months in the residence) aged ≥3 years completed this survey. Individuals with bivalve shellfish consumption data who were screened according to the food codes of bivalve shellfish in the “Chinese Food Composition Table Standard Edition [56]” were known as consumers only. In total, 1083 bivalve shellfish consumers were selected for the present study. After excluding people with missing information (such as age, weight, and shellfish consumption) and those with extreme values, 1075 individuals who consumed a variety of bivalve shellfish, including mussels, oysters, scallops, clams, *Scapharca subcrenata* and so on were finally included. All the participants signed an informed consent form, and their personal information was kept confidential.

Owing to the acute toxic effects of PSTs, it is important to determine a large portion size rather than long-term average consumption to safeguard the health of the consumers [4]. Therefore, upper- and lower-percentile food consumption amounts should be defined based on individual consumer days. For surveys collecting data on multiple consumption days per person, the individual consumer days were assumed to be independent observations in the derivation of the upper and lower percentiles [39]; that is, these valid consumer days were considered to be independent observations in the database, and were not averaged when calculating the percentile of consumption.

The consumption and body weight data consisted of five age groups [57]: young children (≤6 years), older children (7–13 years), adolescents (14–17 years), adults (≥18 years), and older adults (≥60 years).

### 5.7. Assessment Methods

Point-estimate modeling was performed to assess the dietary exposure of the Zhejiang consumers to PSTs. The dietary exposure to PSTs was calculated according to the following equation for each age group [20,39]:(3)$$PSTs\;dietaryexposure\;(µg\;STX\;eq.{kg}^{-1}\;bw\;{day}^{-1})=\frac{Concentration\;of\;PSTs\;in\;shellfish\;\left(µg\;STX\;eq.\cdot {kg}^{-1}\right)\;\times \;Shellfish\;consumption(g\cdot d^{-1})}{Body\;weight(kg)}$$

The potential health risks of PSTs were calculated by dividing the output of the exposure estimates of the PSTs in shellfish by the corresponding ARfDs. In the present study, the ARfD for PSTs was 0.5 µg STX eq./kg b.w., as recommended by EFSA [4], and 0.7 µg STX eq./kg b.w., as recommended by the FAO/WHO/IOC [21]. It was assumed to be safe when the percentage value of ARfD (%ARfD) was ≤100%; conversely, the risk was considered to be unacceptable [58].

Owing to differences in shellfish intake among individuals and varying levels of PST contamination in the samples, PST intake was classified according to four consumption scenarios. Assessments were performed according to the different age groups. Considering the extreme values of the consumption data obtained from the questionnaire survey, the 95th percentile of daily consumption was used to represent a large portion size [4]. The concentrations were derived using UPLC-MS/MS and were relatively accurate. Therefore, the maximum concentration indicated a high contamination level. The four consumption scenarios were as follows: Scenario A: median consumption and concentration; Scenario B: median consumption and maximum concentration; Scenario C: the 95th percentile of daily consumption and median concentration; and Scenario D (acute exposure): the 95th percentile of daily consumption and maximum concentration.

### 5.8. Statistical Analysis

All the statistical analyses were performed using IBM SPSS 25.0 (IBM Corp, Armonk, NY, USA). Measurement data without a normal distribution were presented as the median and 95th percentile. Enumeration data were presented as rates. Chi-square and Fisher’s exact tests were used to determine whether the detection rates of the PSTs in the shellfish depended on the sampling species, site, and time. Bonferroni correction was applied for multiple testing of partially correlated measurements. Statistical significance was set at *p* < 0.05.

## Figures and Tables

**Figure 1 toxins-15-00624-f001:**
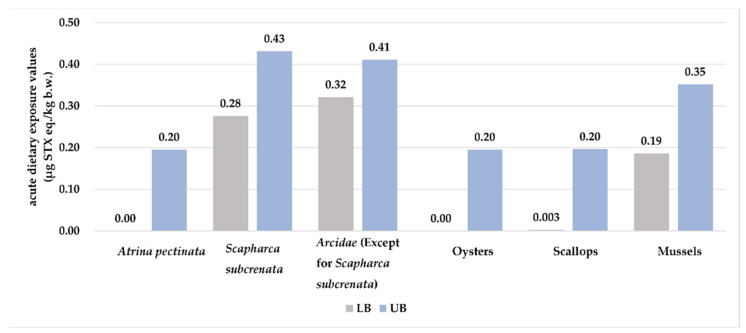
Acute dietary exposure to PSTs from each shellfish species in the general population of consumers only. LB: ND = 0; UB: ND = LOD.

**Table 1 toxins-15-00624-t001:** Occurrence and raw concentration ranges of ten individual PSTs in samples (n = 546).

Individual PSTs	N (%) ^1^	Concentration Range
		(<LOD ^2^-Max, µg/kg)
STX	13 (2.4)	<20–168
neoSTX	4 (0.7)	<20–101
GTX1	0 (0)	ND ^3^
GTX2	8 (1.5)	<20–128
GTX3	4 (0.7)	<20–60.2
GTX4	0 (0)	ND
GTX5	23 (4.2)	<20–242
dcSTX	1 (0.2)	<20–26
dcGTX2	13 (2.4)	<20–58.4
dcGTX3	3 (0.5)	<20–23.1

^1^ N (%), the number and percentage of samples above LOD. ^2^ LOD, limit of detection. ^3^ ND, Non-detected.

**Table 2 toxins-15-00624-t002:** Detection rates and concentrations of PSTs in samples.

		Sample Size	Detected N (%) ^1^	Mean ^2^		Median ^3^		Max ^4^	
				(µg STX eq./kg)		(µg STX eq./kg)		(µg STX eq./kg)	
				LB ^5^	UB ^6^	LB	UB	LB	UB
	Total species	546	38 (7.0)	4.1	135.1	0	132.2	186.9	291.9
Species									
	*Atrina pectinata*	2	0 (0)	0	132.2	0	132.2	0	132.2
	*Scapharca subcrenata*	24	10 (41.7)	53.7	174.2	0	132.2	186.9	291.9
	*Arcidae* (Except for *Scapharca subcrenata*)	53	3 (5.7)	8.5	137.3	0	132.2	217.4	278.2
	Oysters	77	0 (0)	0	132.2	0	132.2	0	132.2
	Scallops	36	1 (2.8)	0.1	132.2	0	132.2	1.9	132.9
	Mussels	354	24 (6.8)	1.4	133.0	0	132.2	126.0	238.2
Sampling time									
May	2018	30	0 (0)	0	132.2	0	132.2	0	132.2
	2019	70	8 (11.4)	4.7	135.0	0	132.2	126.0	238.2
June	2018	123	3 (2.4)	0.1	132.3	0	132.2	5.4	136.4
	2019	134	13 (9.7)	1.3	132.8	0	132.2	35.6	153.6
July	2018	40	0 (0)	0	132.2	0	132.2	0	132.2
	2019	28	6 (21.4)	32.2	159.3	0	132.2	186.9	291.9
August	2018	40	0 (0)	0	132.2	0	132.2	0	132.2
	2019	10	0 (0)	0	132.2	0	132.2	0	132.2
September	2018	48	2 (4.2)	3.7	135.0	0	132.2	176.3	267.3
	2019	23	6 (26.1)	28.8	148.8	0	132.2	217.4	278.2
Sampling site									
	Hangzhou	29	12 (41.4)	5.7	134.8	0	132.2	35.6	153.6
	Ningbo	124	3 (2.4)	0.1	132.3	0	132.2	5.4	136.4
	Taizhou	124	16 (12.9)	8.5	137.1	0	132.2	217.4	278.2
	Wenzhou	145	0 (0)	0	132.2	0	132.2	0	132.2
	Zhoushan	124	7 (5.6)	8.3	139.2	0	132.2	186.9	291.9

^1^ N (%), the number and percentage of samples contaminated with PSTs; ^2^ Mean, arithmetic mean; ^3^ Median, the 50th percentile; ^4^ Max, the maximum value in a group; ^5^ LB, ND = 0; ^6^ UB, ND = LOD.

**Table 3 toxins-15-00624-t003:** Dietary exposure to PSTs in the general population and different age groups of consumers only according to four scenarios.

Age (Years)	N	Body Weight (kg)	Consumption(g/d)		Dietary Exposure LB ^1^-UB ^2^ (µg STX eq./kg b.w.)				%ARfD ^3^ LB-UB (%)			
			Median	P95	A ^4^	B ^5^	C ^6^	D ^7^	A	B	C	D
All	1075	57.9	30.0	85.5	0–0.07	0.10–0.15	0–0.20	0.28–0.43	0–14 (0–10)	20–30 (14–21)	0–40 (0–29)	56–86 (40–61)
≤6	50	19.2	19.5	64.0	0–0.13	0.19–0.30	0–0.44	0.62–0.97	0–26 (0–19)	38–60 (27–43)	0–88 (0–63)	**124**–**194** (89–**139**)
7–13	83	35.4	23.4	58.5	0–0.09	0.12–0.19	0–0.22	0.31–0.48	0–18 (0–13)	24–38 (17–27)	0–44 (0–31)	62–96 (44–69)
14–17	22	57.7	31.2	78.9	0–0.07	0.10–0.16	0–0.18	0.26–0.40	0–14 (0–10)	20–32 (14–23)	0–36 (0–26)	52–80 (37–57)
18–59	817	62.0	30.0	92.0	0–0.06	0.09–0.14	0–0.20	0.28–0.43	0–12 (0–9)	18–28 (13–20)	0–40 (0–29)	56–86 (40–61)
≥60	103	62.2	27.6	71.0	0–0.06	0.08–0.13	0–0.15	0.21–0.33	0–12 (0–9)	16–26 (11–19)	0–30 (0–21)	42–66 (30–47)

^1^ LB, ND = 0. ^2^ UB, ND = LOD. ^3^ %ARfD, the percentage of exposure to the ARfDs recommend by EFSA and FAO/WHO/IOC (in parentheses), with >100% in bold. ^4^ A, Scenario A: median shellfish consumption and PST concentration (0(LB)–132.2(UB) µg STX eq./kg). ^5^ B, Scenario B: median shellfish consumption and maximum PST concentration (186.9 (LB)–291.9 (UB) µg STX eq./kg). ^6^ C, Scenario C: the 95th percentile (P95) daily shellfish consumption and median PST concentration.^7^ D, Scenario D (acute exposure): the 95th percentile (P95) daily shellfish consumption and maximum PST concentration.

**Table 4 toxins-15-00624-t004:** One-time safe serving size of each shellfish species for young children (Regarding edible portions).

Species	Contamination Level ^1^ (µg STX eq./kg)	Serving Size 1 ^2^(g/d)	Serving Size 2 ^3^(g/d)
*Atrina pectinata*	132.2	72.8	101.9
*Scapharca subcrenata*	291.9	33.0	46.1
*Arcidae* (Except for *Scapharca subcrenata*)	278.2	34.6	48.4
Oysters	132.2	72.8	101.9
Scallops	132.9	72.4	101.3
Mussels	238.2	40.4	56.5

^1^ Contamination level, the upper bounds (UB) of maximum PST concentration in each species. UB: ND = LOD. ^2^ Serving size 1 was based on the ARfD recommended by EFSA (0.5 µg STX eq./kg b.w.).^3^ Serving size 2 was based on the ARfD recommended by FAO/WHO/IOC (0.7 µg STX eq./kg b.w.).

## Data Availability

The data presented in this study are available upon request from the corresponding author due to restrictions, e.g., privacy or ethical.

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
