# Peer review of "An Occurrence and Exposure Assessment of Paralytic Shellfish Toxins from Shellfish in Zhejiang Province, China"

_toxins, 2023, doi:10.3390/toxins15110624_

Round 1

Reviewer 1 Report

The manuscript is very well written and I find it very interesting, I would just like to make one observation.

The manuscript clearly demonstrates the importance of a long-term study of marine biotoxins in the target area.  In addition, the data collected are very well presented to demonstrate the situation in the area.

Although the individual contents of the samples do not pose a risk according to the legislation, the authors assume the different consumer profiles in order to calculate and demonstrate the real risk.

In this way, it can be seen that the children's age group may be the most affected.

A group of 10 paralytic toxins (STX, dcSTX, neoSTX, GTX1, GTX2, GTX3, GTX4, dcGTX2, dcGTX3, and GTX5) were selected for analysis, showing that by monitoring for these toxins alone, shellfish can pose a risk to children.

Is there a possibility to consider non-targeted testing for other PSP toxin congeners? In this way, we could have clearer evidence of total toxicity. 

I believe that this type of study should be carried out with a targeted and non-targeted methodology to try to get an overall view of the contamination of the sample, so I think this is an aspect of the work that needs to be improved. 

The paper is very comprehensive and very well written and therefore I think it should be published.

Author Response

Dear reviewer:

We sincerely thank you for taking the time out of your busy schedule to review our thesis in detail, which has greatly improved the quality of the paper!

We have written the details of these issues in the attached document.

Reviewer 2 Report

The research addresses an important issue for public health,the exposure to PSTs due to consumption of mollusks. The authors show evidence of differences in toxins accumulated by  different species of bivalves.
Despite the evidence shown, it is difficult to determine if the presence of different PST analogues, due to the biotransformation of these toxins in Gymnodinium catenatum, other species that produce these toxins are present and consumed by mollusks, without detecting a bloom of these dinoflagellates
This research shows the importance of monitoring programs to understand the effects of PSTs when these are accumulated and tranfer to humans that consume shellfish  and can be exposed to pasive accumulation for long time
This research shows the importance of monitoring programs to understand the effects of PSTs when these are accumulated in shellfish and tranfer to humans that consume shellfish, and  can be exposed to pasive accumulation for long time, without know their effects on  adults and children

My comments and suggestions wer included in PDF manuscript

Author Response

Dear reviewer:

We would like to express our special thanks to you for your critical comments and suggestions on our paper, which have greatly improved the quality of the paper!

We have written the details of these issues in the attached document.

Reviewer 3 Report

  1. The abstract section requires revision. Initially, the author should address the identified issue and then provide specific information.

  2. The choice of keywords is inappropriate, with a mix of similar words.

  3. This sentence should be more precise: "It poses a threat to public health and results in economic losses."

  4. Numerous sentences lack proper referencing.

  5. It remains unclear which shellfish species were targeted and provided to the population.

  6. There are significant issues with the materials and methods section, which lacks an adequate explanation of the sampling procedure.

  7. The conclusion section should be rewritten. Authors are advised to avoid exaggeration and ensure it aligns with the original aim of the experiment.

  8. Authors are instructed to clearly state the aim/objective of the manuscript.

Author Response

Dear reviewer:

We sincerely thank you for your careful review and guidance, which has led to a great improvement in the quality of our thesis.

We have written the details of these issues in the attached document.

Round 2

Reviewer 3 Report

The manuscript is extensively revised. 

Author Response

Dear reviewer:

We sincerely thank you for taking the time out of your busy schedule to review our paper in detail and for approving our revised manuscript!